# Detection of Parkinson’s Disease from 3T T1 Weighted MRI Scans Using 3D Convolutional Neural Network

**DOI:** 10.3390/diagnostics10060402

**Published:** 2020-06-12

**Authors:** Sabyasachi Chakraborty, Satyabrata Aich, Hee-Cheol Kim

**Affiliations:** 1Department of Computer Engineering, Inje University, Gimhae 50834, Korea; c.sabyasachi99@gmail.com; 2Institute of Digital Anti-Aging Healthcare, Inje University, Gimhae 50834, Korea; 3Ubiquitous Healthcare & Anti-aging Research Center (u-HARC), Inje University, Gimhae 50834, Korea

**Keywords:** Parkinson’s Disease, neurodegeneration, magnetic resonance imaging (MRI), convolutional neural network (CNN), deep learning

## Abstract

Parkinson’s Disease is a neurodegenerative disease that affects the aging population and is caused by a progressive loss of dopaminergic neurons in the substantia nigra pars compacta (SNc). With the onset of the disease, the patients suffer from mobility disorders such as tremors, bradykinesia, impairment of posture and balance, etc., and it progressively worsens in the due course of time. Additionally, as there is an exponential growth of the aging population in the world the number of people suffering from Parkinson’s Disease is increasing and it levies a huge economic burden on governments. However, until now no therapeutic method has been discovered for completely eradicating the disease from a person’s body after it’s onset. Therefore, the early detection of Parkinson’s Disease is of paramount importance to tackle the progressive loss of dopaminergic neurons in patients to serve them with a better life. In this study, 3T T1-weighted MRI scans were acquired from the Parkinson’s Progression Markers Initiative (PPMI) database of 406 subjects from baseline visit, where 203 were healthy and 203 were suffering from Parkinson’s Disease. Following data pre-processing, a 3D convolutional neural network (CNN) architecture was developed for learning the intricate patterns in the Magnetic Resonance Imaging (MRI) scans for the detection of Parkinson’s Disease. In the end, it was observed that the developed 3D CNN model performed superiorly by completely aligning with the hypothesis of the study and plotted an overall accuracy of 95.29%, average recall of 0.943, average precision of 0.927, average specificity of 0.9430, f1-score of 0.936, and Receiver Operating Characteristic—Area Under Curve (ROC-AUC) score of 0.98 for both the classes respectively.

## 1. Introduction

The second most common neurological disorder that prevails among the aging population in the world is considered to be Parkinson’s Disease (PD). Parkinson’s Disease primarily affects the nerve cells in the brain that are responsible for producing dopamine, which is an organic chemical that acts as a neurotransmitter to transmit signals between nerve cells. Therefore, patients having Parkinson’s Disease usually suffer from cognitive and movement disorders such as muscle stiffness, tremors, and impairment of posture and balance. Moreover, Parkinson’s Disease is progressive in nature, therefore, early detection and monitoring of the disease leads to improvement in the life of the patients. Also, as the aging population across the globe in increasing exponentially, a requirement for the development of suitable methods for the detection of Parkinson’s Disease at a very early stage is indeed very important [1,2,3,4,5].

For the early detection of Parkinson’s Disease, the most widely used diagnostic paradigm is the analysis of Magnetic Resonance Imaging (MRI) scans of the brain. The MRI scans provide anatomical details about the subcortical structures of the brain that are further analyzed to check for any aneurysms, which further deems helpful for the early diagnosis of a particular type of disease. However, as the MRI is a 3D structure, it becomes really difficult for the human eye to analyze the intrinsic details and heterogeneous properties of subcortical structures [6,7]. Therefore, with the advancement of intelligent technologies, computer-aided detection systems have proven to be very effective in performing analysis and diagnosis of diseases by leveraging multi-dimensional healthcare data.

In the past researchers performed multiple studies and found out that the textural and morphological analysis of the tissue and cell imaging scans have provided some very astonishing results. The application of textural and morphological analysis was considered to be huge as it was able to perform the quantification of grey level patterns and derive the inter-pixel relationship within the regions of interest. Moreover, it was also observed that different areas in a scan or an image had different textural and morphological patterns which were difficult for human beings to track [8]. Therefore, textural and morphological analysis of the imaging scans proved to be very much reliable for neurological studies and applications in the detection and diagnosis of diseases. But with the advances in the field of computer applications and intelligent systems, the research community is now focusing more on data-driven feature representation rather than handcrafted feature engineering which requires domain-specific knowledge [9]. Therefore, with the rapid development of deep learning architectures and technologies, it is proving to lay down some state-of-art methodologies for medical image applications.

The paper is structured as follows. The related works in the field of the detection of Parkinson’s Disease and other Neurodegenerative diseases using deep learning and machine learning algorithms are described in Section 2. Section 3 discusses the data used in the study specifically, the data selection procedure, imaging modalities, and image registration protocols. Moreover, the section demonstrates the complete workflow for the processing of the Parkinson’s Progression Markers Initiative (PPMI) data and registering the data with respect to MNIPD25-T1MPRAGE-1 mm atlas. Section 4 underlines the methodology of the study concerning the development and optimization of the 3D convolutional neural network architecture. Section 5 discusses the performance and generalizing capability of the developed 3D convolutional neural network architecture. Section 6 further discusses the whole study and discusses the importance of the work for the early detection of Parkinson’s Disease. Finally, the paper is concluded in Section 7 and the future work regarding the study is discussed.

## 2. Related Work

In recent years the use of deep learning for image analysis has presented some of the most astounding results. More advertently in the field of medical image analysis such as classification, segmentation, etc. [10,11,12], convolutional neural networks (CNN) [13] had been considered to demonstrate some state-of-the-art results and also are used in many real-world applications. In the field of Alzheimer’s disease (AD) detection, multiple notable works have been performed and presented some state-of-the-art results. Payan and Montana [14] performed a study where 2265 historical MRI scans were obtained from the Alzheimer’s Disease Neuroimaging Initiative (ADNI) and further all the MRI scans were subjected to a sparse-autoencoder followed by a 3D CNN network. The classification of AD vs. healthy control (HC) vs. mild cognitive impairment (MCI) was plotted at an accuracy of 89.47%. Also, Hosseini et al. [15] undertook a study for the classification of AD, HC, and MCI by leveraging 3D convolutional autoencoder on normalized T1 weighted MRI scans from the ADNI database. Similarly, others have also demonstrated their works where CNN was leveraged in MRI and Functional magnetic resonance imaging (fMRI) scans for the detection of AD and the classification of MCI and HC [16,17,18,19,20].

On the other hand, for the early detection of Parkinson’s Disease, multiple works have been performed considering the texture and morphological analysis of the region of interest (ROI) and voxels of interest (VOI) of subcortical structures of the brain. Li et al. [21] performed a study regarding the 3D texture analysis of the substantia nigra using the quantitative susceptibility maps (QSM) and R2* maps for the detection of Parkinson’s Disease. Further, the study extracted first- and second-order textural features from QSM and R2* maps and it was found that first- and the second-order QSM maps accurately distinguished PD from HC. Similarly, Sikio et al. [22] proposed a technique to determine the structural changes of the brain from an MRI baseline using textural features. It was observed from the study that the textural features can be considered for discriminating the structural changes. Moreover, similar studies were also performed by [4,23] where the textural and morphological features were only considered from specific subcortical structures of the brain for the prediction of Parkinson’s Disease. But concerning the implementation of CNN and deep learning on MRI scans for the detection of Parkinson’s Disease, there are quite a few notable works that have been performed. Therefore, following in this section some related studies are discussed that have been considered to demonstrate state of the art results.

Ortiz et al. [24] performed a study for the detection of Parkinson’s Disease using features based on the isosurface of 3D brain Single-Photon Emission Computed Tomography (SPECT) scans. For the study, the authors acquired the DaTscan SPECT scans from PPMI database. The SPECT scans were further subjected to a feature extraction method that extracted only the isosurface or isolines (2D version of isosurfaces) from the 3D SPECT scans. Further, the isosurfaces were subjected to a 3D CNN model based on the characteristics of AlexNet and it was observed that the model plotted a specificity and sensitivity of 95% and receiver operating characteristic (ROC) of 0.97. Similarly, Kollia et al. [25] also performed an interesting study where they predicted Parkinson’s Disease from MRI and DaT scan data by leveraging latent information from deep convolutional neural networks. In the study, the authors implemented two deep neural networks, one with the normal specification of convolution layers followed by fully connected layers and the other one where the normal architecture was followed by a recurrent layer of gated recurrent unit (GRU) neurons. Further, latent variables were extracted from the fully connected layer of the trained CNN models and multiple approaches for the detection were performed. The approaches included combinations of transfer learning, clustering [26], and nearest neighbors for enhancing the classification performance of the model.

Shinde et al. [27] in his work regarding the detection of Parkinson’s Disease leveraged neuromelanin sensitive MRI (NMS-MRI), which can identify abnormalities in the substantia nigra (SNc) in Parkinson’s Disease patients. Moreover, the author also plotted concerns over the handcrafted features based on the contrast ratio, area, and volumes of the subcortical structures [28,29,30,31]. Further, the authors employed a CNN network for the detection of PD from HC and demonstrated optimum test accuracy of 80% concerning ratio-based and radiomics feature-based classification techniques. In another study, Sivaranjini et al. [32] leveraged AlexNet [33] architecture on MRI images obtained from the PPMI database for the detection of Parkinson’s Disease. The model provided an optimum accuracy of 88.9% while distinguishing PD from HC. Similar studies were further performed by [34,35,36] by leveraging multiple imaging modalities, multiple imaging cohorts, and different types of deep learning architecture. The studies further performed superiorly in identifying Parkinson’s Disease from healthy control.

The aforementioned literature further proves the diligence of the analysis of MRI by leveraging deep learning architectures for the detection of Parkinson’s Disease. Moreover, it has also been established that deep learning architectures and more specifically convolutional neural networks perform superiorly in extracting hierarchical information from imaging modalities. Therefore, the findings plotted by the aforementioned research studies motivated us to perform an analysis of the MRI scans using a 3D CNN network for the detection of Parkinson’s Disease.

## 3. Data Collection and Preprocessing

### 3.1. Data Collection

The data for the study was collected from the PPMI database (www.ppmi-info.org/data). PPMI database for neuroimages is considered to be a landmark, international, and multicenter study to research the biomarker’s that are responsible for Parkinson’s Disease progression. The MRI scans selected for the study were based on particular imaging protocols described in Table 1 and also corresponds to the baseline visit. Further, all the scans that were considered in the study were obtained from a single type of scanner, i.e., Siemens, Munich, Germany. Moreover, all the acquired scans were based on Magnetization Prepared—Rapid Gradient Echo (MP-RAGE) sequence. All the scans used in the study were acquired in a time range of 20–30 min field of view (FoV) of all the scans including vertex, cerebellum, and pons.

Post applying the filter based on the imaging protocol mentioned in Table 1, a total of 406 MRI scans were selected from the baseline visit of the patients. Out of 406 patients, 148 were female and 258 were male. The scans that were considered for the study belonged to the subjects aged 62.64 ± 9.9. The scans primarily belonged to two research groups that are healthy control (HC) and Parkinson’s Disease (PD). The scans were distributed into the respective research groups as 203 for healthy control and 203 for Parkinson’s Disease. The subjects who were considered for obtaining the scan were selected on certain criteria described in Table 2.

Table 3 shows the specifications of the scans that were obtained from the PPMI database. While Figure 1 depicted shows the sample of MRI scans belonging to both the research groups that were obtained from the PPMI database.

### 3.2. Data Preprocessing

The dataset that was used in the study was obtained from the PPMI database; as previously mentioned, PPMI is a multicenter study, therefore, the imaging scans acquired in the study contained temporal and spatial differences. To solve this particular problem and to maintain a constant modality between all the scans it was required that all the scans needed to be in the same space such as Montreal Neurological Institute (MNI) [37,38] or individual brain atlases using statistical parametric mapping (IBASPM) [39]. Therefore, to transform the PPMI MRI data that has been collected from multiple centers across the globe to a fixed coordinate system, an image registration procedure was performed. Image registration is a process where traversal is performed on a fixed image (atlas) to find the alignment parameters and coordinates so that an unknown or an unseen image can be aligned similarly to the fixed image. Trivially, image registration could be understood as the process of aligning two images to a particular space where one acts as the source image and the other as target image, and the source image is transformed in a method to align with the target image. In the specific study, the MRI scans obtained from the PPMI database were considered as the source image and the atlas, such as MNI or IBASPM, were considered as the target image.

The registration of the MRI scans obtained from the PPMI database was performed using MNIPD25-T1MPRAGE-1 mm atlas created by [40,41,42]. The specifications of the MNIPD25-T1MPRAGE-1 mm atlas is described in Table 4. The registration of the MRI scans was performed using one of the most effective normalization tools known as advanced normalization tools python (ANTsPy) [43]. ANTsPy is used particularly in the field of imaging research for extracting important information from complex imaging datasets to perform preprocessing on MRI, fMRI, and SPECT data. The registration of the acquired MRI scans with the MNIPD25-T1MPRAGE-1 mm atlas was performed using symmetric normalization. Figure 2 depicts a particular MRI scan before and after the registration process.

## 4. Materials and Methods

The main premise of the study focusses on the detection of Parkinson’s Disease and the classification of MRI scans as healthy control or Parkinson’s Disease using 3D convolutional neural networks. The complete flow of process and methodology for the detection of Parkinson’s Disease is described in Figure 3. The methodology was primarily divided in to four stages: MRI scan acquisition from the PPMI database; data preprocessing, registration, and transformation; 3D convolutional neural network architecture; and finally the results and performance evaluation of the CNN architecture based on some metrices. The first two stages of the methodology have been thoroughly discussed in Section 3 and the third and the fourth stage will be discussed in the following sections.

### 4.1. 3D Convolutional Neural Network Architecture

In recent times, supervised learning techniques for solving problems have evolved massively. Moreover, the popularity and effectiveness of deep learning algorithms have also undergone a major paradigm shift in terms of architectural designs and optimizer functions [43]. Particularly, in the field of health care, deep learning algorithms have shown much predominance over the previous techniques that were used for imaging analysis, aneurysms detection in images, biosignal analysis, etc.

In this paper, a 3D convolutional neural network model has been developed for the detection of Parkinson’s Disease from T1 weighted MRI scans. The primary proposition of the work presents a system that can be used to identify Parkinson’s Disease from MRI or brain images. Additionally, the second proposition of the study was to determine the plausible regions of interest (ROIs) in the brain MRI images that are responsible for Parkinson’s Disease. Therefore, to solve the primary proposition of the study a 3D convolutional neural network has been developed as shown in Figure 4 and Table 5. The CNN network developed in the work consists of 35 layers including the input and the output layer. Further, the network architecture consists of 12 3D convolution layers, which allows the model to create the feature representations of the input brain MRI scans. Moreover, all the convolution layers are supported by activation functions. Further, all the feature representations are subjected to max-pooling layers which are responsible for down-sampling the input feature matrix and provides an abstract form of the feature representation to avoid overfitting. After the complete process of feature learning, all the feature matrices are flattened so that it can be accepted by the dense layer or the fully connected layer. The representations from the dense layer are further subjected to the output dense layer with two neurons and sigmoid activation which corresponds to the two states that are healthy control and Parkinson’s Disease.

### 4.2. Hypothesis and Training Procedure

For developing the statistical, machine learning, and deep learning model, the first step is considered to be the development of the hypothesis of the problem that needs to be solved. Therefore, the primary hypothesis that was devised for solving a particular problem is as follows:The recall of the Parkinson’s Disease class or the true negatives must be more than 0.95 and the probability factor of the mispredicted samples or the false negatives must not be more than 60%.The recall of the healthy class or the true positives must be more than 85% and the probability factor of the mispredicted samples or the false positives must not be more than 65%.

Therefore, based on the above hypotheses, the performance of the 3D convolutional neural network model was evaluated. For the evaluation purpose, five different classification performance metrics were considered namely accuracy, precision, recall, *f1*-score, and confusion matrix. Also, for determining the generalizability of the model over unseen data a 5-split cross-validation was performed. The details regarding the evaluation of the performance metrics are described in the Results section.

### 4.3. Model Optimizer Hyperparameters and Loss

The development of the 3D CNN architecture is indeed the most important aspect of the work. However, the component that needs to be considered carefully for creating the learning algorithm is choosing the right set of hyperparameters for optimizing the internal set of parameters of the network such as weights and biases, and the loss function.

The process of controlling the training process is considered very important while creating a deep learning model. The process is undertaken by the hyperparameters of the optimizer function that is responsible for tuning the optimizer algorithms. For the present study, the primary aspect that lies in the optimization algorithm is to minimize the validation and the testing error of the model. For performing the specific task, the hyperparameters that reside outside the primary deep learning model must be tuned in such a way we generate the perfect internal parameters of the model that are the weights and biases. However, the challenge that is faced in the process is that the hyperparameters much be chosen in a particular way that it should be model-specific rather than a training set to increase the generalizability of the model over unseen data. Therefore, for choosing the perfect set of hyperparameters to maintain the overall model generalizability and optimum objective score, Bayesian sequential model-based optimization (SMBO) is used.

Bayesian SMBO is an algorithm used for hyperparameter optimization the works to minimize an objective function by creating a surrogate model (probability function) based on the evaluation results of the previous objective function. The basic objective function of the Bayesian SMBO is given as:(1)$$P\left(score|hyperparameters\right)=\frac{P(hyperparameters\;|score)P\left(score\right)}{P\left(hyperparameters\right)}$$

The surrogate model that is developed by the Bayesian SMBO is considered to be less expensive than the main optimizer function [44]. Further, the next set of evaluation results are selected by using the expected improvement criterion [45]. The criterion is defined as:(2)$$EI\left(x\right)=E\left(\max\left(f\left(x\right)-f^{*},\;0\right)\right)$$
where *x* belongs to the hyperparameter values and is considered to be an improvement in the objective sore of *f(x)*, and *f** is the maximum value of the objective score found in the process.

Further, in the process, AdaDelta [46] is chosen as the optimizer algorithm for optimizing the weights and biases of the network. AdaDelta is considered to be a very robust algorithm relating to the gradient descent algorithm. The algorithm dynamically adapts over the course of the training process by leveraging only first-order information. Moreover, the algorithm does not require any manual tuning of the learning rate and is very robust towards noisy gradient information. Therefore, Bayesian SMBO was applied to the algorithm to generate the optimum hyperparameters and is mentioned below.
*Learning rate: 0.08423; rho: 0.625; epsilon: 1.0*

Another very integral part of the deep learning models is the loss functions. These functions are typically used to determine the variability between the prediction $\left(\hat{y}\right)$ and the true value (y). The output of the loss functions is a non-negative value that increases the generalizability of the model by decreasing the loss [47]. The loss function of a model is given by:(3)$$L\left(\theta \right)=\frac{1}{n}\overset{n}{\underset{i=1}{\sum }}L(y^{\left(i\right)},f\left(x^{\left(i\right)},\theta \right)$$
where θ represents the parameters of the model, *x* represents the feature matrix, and *y* represents the true values for a particular set of features.

The loss function used in the present study was binary cross-entropy, which is also known as the cross-entropy or the log loss. In the binary cross-entropy loss, each prediction outcome is compared to the true value and a loss score is calculated. The loss score in the process is used to penalize the probability of the prediction. The loss score is logarithmic which means a small penalty is allotted to tiny differences between prediction outcome and true value and a large penalty is applied to bigger differences [48]. The equation of binary cross-entropy is given where *y* is the true value and *p(y)* is the predicted probability of y being true.
(4)$$H_{p}\left(q\right)=-\frac{1}{N}\overset{N}{\underset{i=1}{\sum }}y_{i}\log\left(p\left(y_{i}\right)\right)+(1-y_{i})\;log\left(1-p\left(y_{i}\right)\right)$$

## 5. Results

The 3D convolutional neural network model presented decent results in terms of detecting Parkinson’s Disease from brain MRI scans. The model quantitatively presented effective results by prompting an average recall and precision of 0.9421 and 0.9280 for both the classes, respectively. Also, for the training procedure, a 5-split cross-validation with the ratio of 80:20 was performed over the complete dataset, and it was observed that the model demonstrated optimum generalizability in terms of predicting the data according to multiple cross-validation test sets. Table 6 shows the results of the 5-split cross-validation that was used to determine the generalizability of the 3D convolutional neural network model over unseen data.

Figure 5 depicts the confusion matrix that was generated based upon the results received from the best performing cross-validation set. Also, it can be observed that the confusion matrix completely aligns with the prior hypothesis which states that there must not be any misprediction of the MRI scans belonging to the PD class into any other class and the recall of prediction in the healthy class must be more than 85%. However, analyzing just the predictive performance of the deep learning models are not sufficient to determine the credibility of the model. Therefore, in Figure 6 and Figure 7 quantile–quantile plots (QQ) have been depicted to understand the uncertainty of the model and the confidence of predictions.

Another very important factor that needs to be measured for evaluating the performance of the deep learning model is the interpretability of the models. The field of healthcare is considered to be a critical field when it comes to the implementation of automated intelligent systems. Therefore, the prime requirement that needs to be provided out of the model is the interpretation behind a particular prediction or cause–effect information that led to a particular prediction. Therefore, to interpret the predictions of the developed 3D CNN model, a class activation map was used [49,50,51]. Figure 8 shows the class activation map on the sample MRI slices that has been predicted as Parkinson’s Disease. The class activation map shows that the model paid much attention to the region of substantia nigra pars compacta (SNc), which is most affected due to the loss of dopaminergic neurons.

## 6. Discussion

The study concerns the development of a 3D convolutional neural network architecture for the detection of Parkinson’s Disease from 3T-T1 weighted MRI scans. MRI scans were collected from the PPMI database from two different research groups namely, healthy control and Parkinson’s Disease. As discussed, the PPMI is a multicenter study, therefore, the acquired MRI scans had spatial and temporal differences. Therefore, to bring all the MRI scans to the same space, an image registration routine was performed over all the MRI scans. The registration of images was performed using MNIPD2-T1MPRAGE-1 mm atlas. Following registration of the brain MRI scans, a 3D convolutional neural network was developed for the learning intricate patterns in the MRI scans for the detection of Parkinson’s Disease and classifying MRI scans into healthy control and Parkinson’s Disease categories, respectively.

Before the development of the model, a hypothesis was designed to evaluate the performance metrics of the model. The hypothesis stated that the recall of the Parkinson’s Disease class or the true negatives must be more than 0.95 and the probability factor of the mispredicted samples or the false negatives must not be more than 60%, and the recall of the healthy class or the true positives must be more than 85% and the probability factor of the mispredicted samples or the false positives must not be more than 65%. Therefore, to satisfy the prior hypothesis five performance metrics namely, confusion matrix, accuracy, precision, recall, and f1-score, were evaluated. From the results, it can be observed that the model predicted superiorly by plotting a maximum accuracy of 95.29%, recall of 0.943, the precision of 0.927, and an f1-score of 0.936 for both the classes. Additionally, for understanding the generalizing capability of the model, 5-split cross-validation was performed and it was observed that the model performed constantly over all the cross-validation splits.

Further to prove the prior hypothesis of the work, a confusion matrix and quantile–quantile plots were depicted in the work, which completely aligned with the hypothesis. In the confusion matrix, it can be observed that there was no misprediction for MRI scans belonging to Parkinson’s Disease class. Moreover, from the quantile–quantile plot, it can be observed that all the correctly predicted samples had the prediction confidence of more than 85% for the MRI scans belonging to the Parkinson’s Disease class. Therefore, it can be observed from the results that the developed 3D CNN model performed robustly by plotting superior performance results by completely aligning with the prior hypothesis of the work, and also the model demonstrated a high generalizing capability based on the cross-validation results.

Presently, as the research area in artificial intelligence, machine learning, and deep learning are focused on the interpretability of the black box models. Therefore, to determine and understand whether the model is choosing the correct areas or the regions that are responsible for the detection of Parkinson’s Disease, a 3D class activation map (3D CAM) was developed. From the 3D CAM plot depicted in Figure 8, it can be observed that the model has predicted a particular MRI scans as Parkinson’s Disease by paying much attention to the substantia nigra region (SNc). Therefore, it can be considered that the model understood the particular areas in the MRI scans which are responsible for Parkinson’s Disease.

## 7. Conclusions

In the proposed study, a 3D MRI analysis was performed for the detection of Parkinson’s Disease using 3D convolutional neural network. The study leveraged full brain 3D MRI scans to understand intricate patterns in all the subcortical structures of the brain for the detection of Parkinson’s Disease. For the evaluation of the CNN model, certain performance metrics were considered, and to validate the values of the performance metrics a prior hypothesis was designed. After, the training of the 3D CNN model it was observed that the model performed superiorly by closely aligning with the prior hypothesis of the study and also demonstrated pretty astounding results. The model developed in the study plotted an overall accuracy of 95.29%, average recall of 0.943, average precision of 0.927, and f1-score of 0.936 for both the classes. Moreover, the interpretation of the model over the MRI scans was also evaluated using 3D class activation maps, and it was found that the model paid maximum attention to the substantia nigra region (SNc) for predicting a particular MRI scan as Parkinson’s Disease.

To conclude, the outcome of the proposed study is very motivating. However, there remains a huge scope of untouched study concerning the development of innovative architectures that can be leveraged for the detection of Parkinson’s Disease using 3D CNN. Moreover, presently the study focused on whole-brain MRI scans, but in the future, it is highly recommended to perform such research by considering specific subcortical structures and the development of more efficient architectures for the detection of Parkinson’s Disease.

## Figures and Tables

**Figure 1 diagnostics-10-00402-f001:**
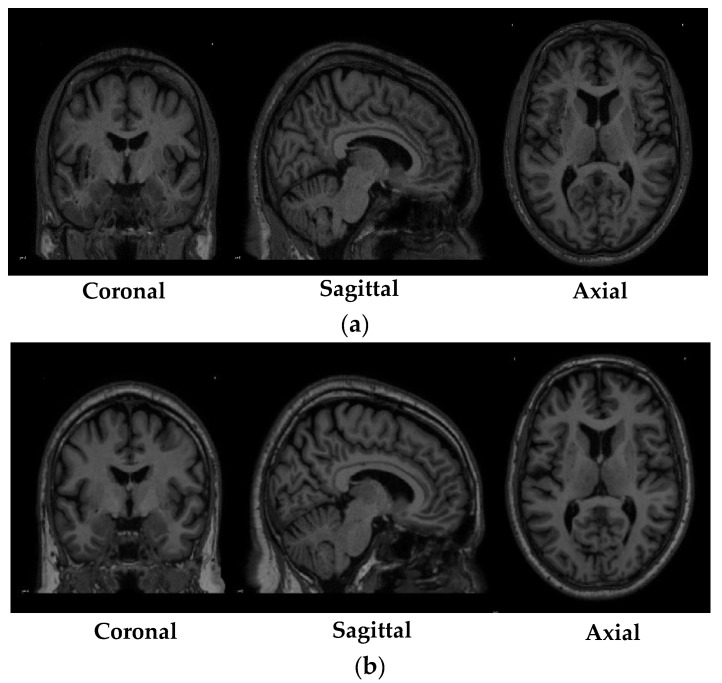
Sample of Magnetic Resonance Imaging (MRI) scans obtained from the Parkinson’s Progression Markers Initiative (PPMI) database (**a**) MRI scan of a subject from the control group; (**b**) MRI scan of a subject from to the Parkinson’s Disease group.

**Figure 2 diagnostics-10-00402-f002:**
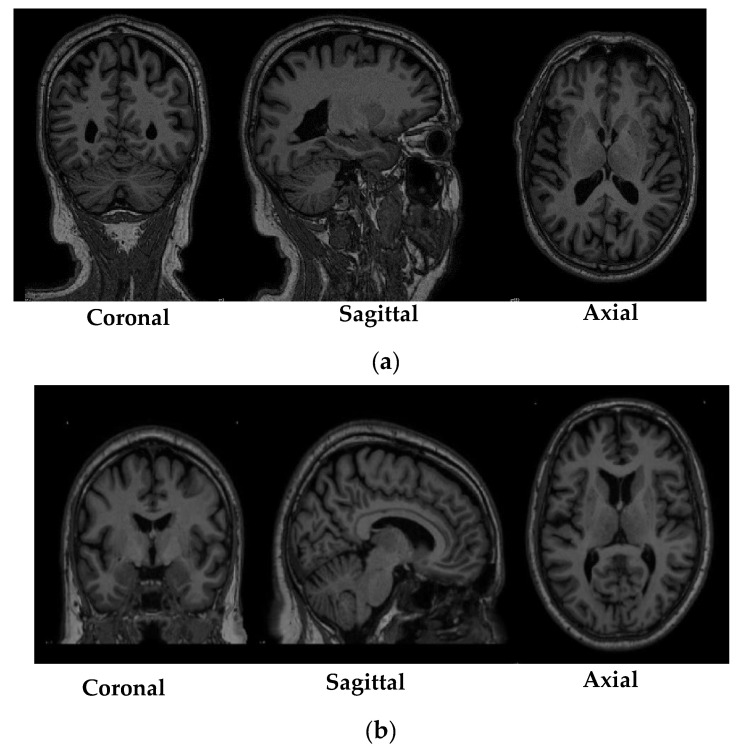
Before and after registration of a particular scan, (**a**) MRI scan before registration and (**b**) MRI scan after registration.

**Figure 3 diagnostics-10-00402-f003:**
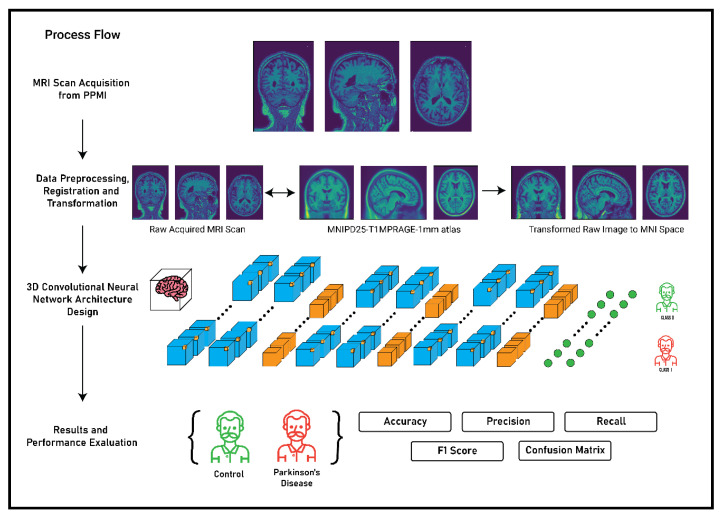
Complete process flow of the study.

**Figure 4 diagnostics-10-00402-f004:**
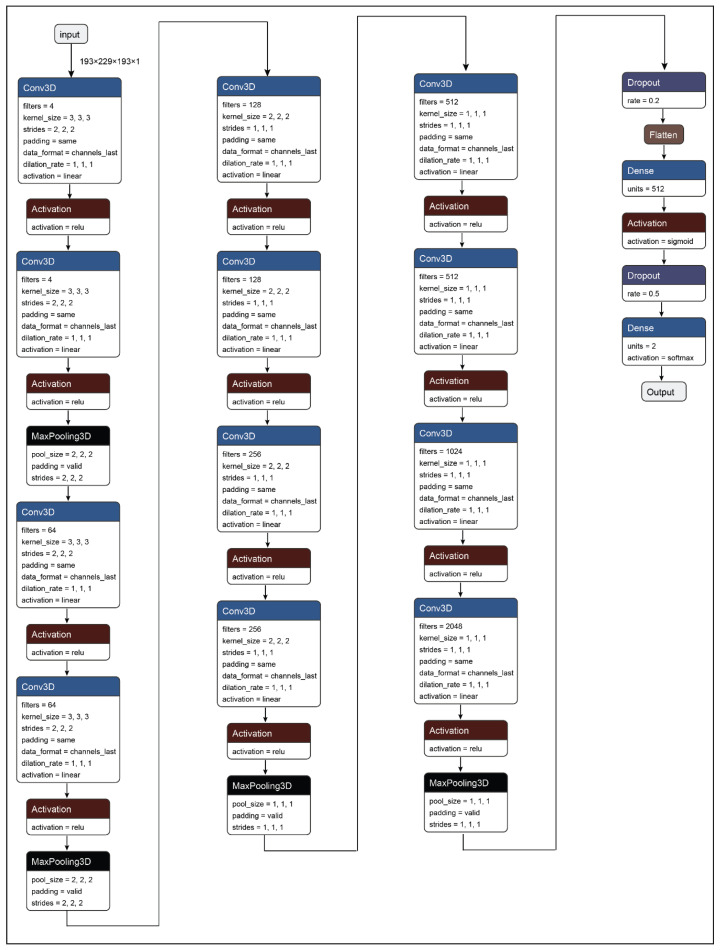
3D Convolutional Neural Network (CNN) architecture.

**Figure 5 diagnostics-10-00402-f005:**
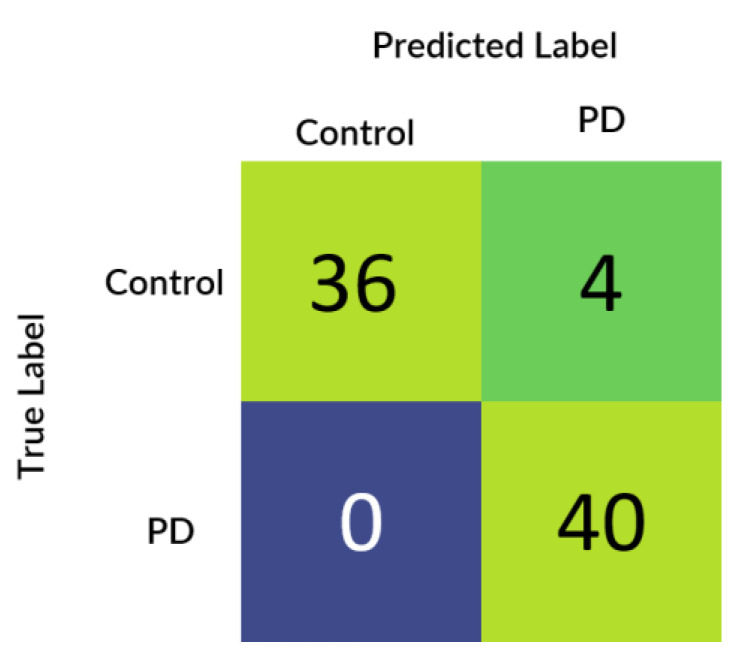
Confusion matrix of the 4th cross-validation split, where PD stands for Parkinson’s Disease.

**Figure 6 diagnostics-10-00402-f006:**
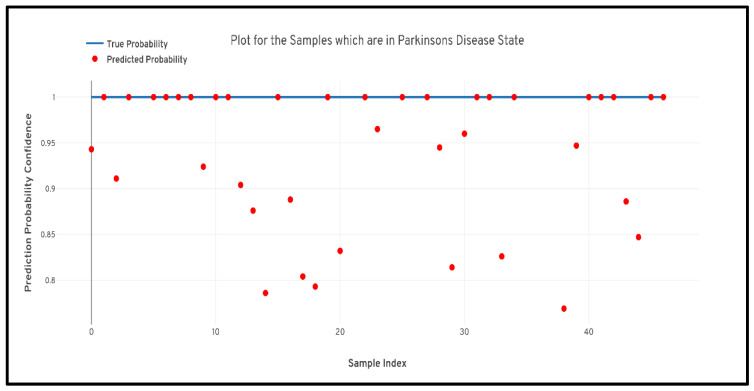
Quantile–quantile plot between the true probability and the prediction probability of the samples belonging to the Parkinson’s Disease class. Probabilistic confidence was 0.932.

**Figure 7 diagnostics-10-00402-f007:**
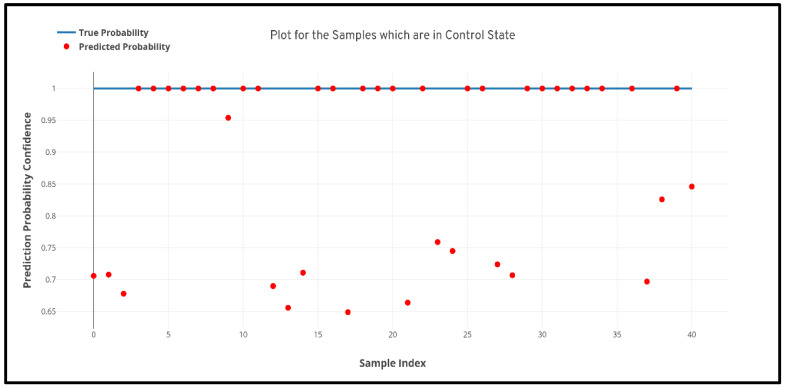
Quantile–quantile plot between the true probability and the prediction probability of the samples belonging to the healthy control class. Probabilistic confidence was 0.843.

**Figure 8 diagnostics-10-00402-f008:**
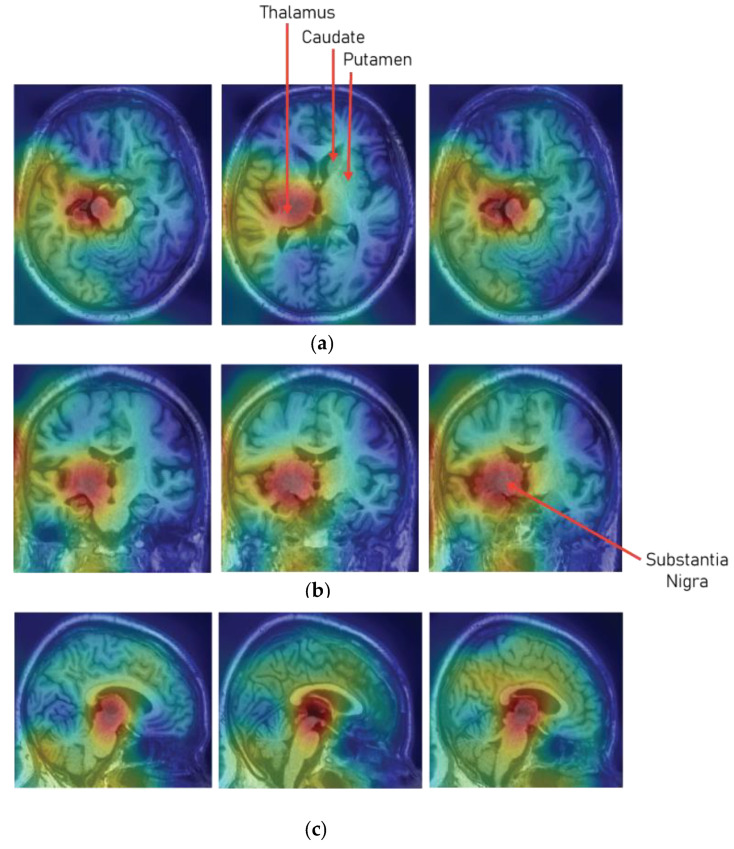
Class activation maps for sample slices of MRI scan that has been predicted as Parkinson’s Disease. (**a**) Axial view, (**b**) coronal view, and (**c**) sagittal view.

**Table 1 diagnostics-10-00402-t001:** Imaging protocol for choosing the scans from Parkinson’s Progression Markers Initiative (PPMI) database.

Imaging Protocol	Values
Modality	Magnetic Resonance Imaging (MRI)
Research Group	Control and Parkinson’s Disease (PD)
Visit	Baseline
Acquisition Plane	Sagittal
Acquisition Type	3D
Field Strength	3.0 Tesla
Flip Angle	9 Degree
Scanner Manufacturing	Siemens Magnetization Prepared—Rapid Gradient Echo (MPRAGE)
Pixel Spacing	0.9–1.5 mm (X &Y)
Slice Thickness	1.0 mm
Weighting	T1

**Table 2 diagnostics-10-00402-t002:** Eligibility criteria for the subject to be included in a research group.

Research Group	Criteria
Parkinson’s Disease	Patients must have at least two: resting tremor, bradykinesia, rigidity, asymmetric resting tremor, asymmetric bradykinesia. Diagnosis of Parkinson’s Disease for two years. Hoehn and Yahr stage I or II. Male or Female age 30 Years or Older.
Control Subjects	Male or Female age 30 Years or Older. Not First degree relative to Parkinson’s patient.

**Table 3 diagnostics-10-00402-t003:** Specification of the acquired scans from PPMI.

Image Parameters	Values
Dimensions	256 × 256 × 170–200 pixels
Interslice Gap	1.0 mm
Slice Thickness	1.0 mm
Spacing	1.0 × 1.0 × 1.0 mm
Plane	Sagittal

**Table 4 diagnostics-10-00402-t004:** Specification of MNIPD25-T1MPRAGE-1 mm atlas.

Image Parameters	Values
Dimensions	193 × 229 × 193 pixels
Interslice Gap	0.0 mm
Slice Thickness	1.0 mm
Spacing	1.0 × 1.0 × 1.0 mm
Plane	Sagittal

**Table 5 diagnostics-10-00402-t005:** 3D convolutional neural network (CNN) architecture.

Layer	Filters	Kernel Size/Pool Size	Stride Size	Output Dimension	Parameters
Convolution_3D_1	4	(3, 3, 3)	(2, 2, 2)	(97, 115, 97, 4)	112
Activation_RELU_1	-	-	-	(97, 115, 97, 4)	0
Convolution_3D_2	4	(3, 3, 3)	(2, 2, 2)	(49, 58, 49, 4)	436
Activation_RELU_2	-	-	-	(49, 58, 49, 4)	0
Max_Pool_3D_1	-	(2, 2, 2)	-	(24, 29, 24, 4)	0
Convolution_3D_3	64	(3, 3, 3)	(2, 2, 2)	(12, 15, 12, 64)	6976
Activation_RELU_3	-	-	-	(12, 15, 12, 64)	0
Convolution_3D_4	64	(3, 3, 3)	(2, 2, 2)	(6, 8, 6, 64)	110656
Activation_RELU_4	-	-	-	(6, 8, 6, 64)	0
Max_Pool_3D_2	-	(2, 2, 2)	-	(3, 4, 3, 64)	0
Convolution_3D_5	128	(2, 2, 2)	(1, 1, 1)	(3, 4, 3, 128)	65664
Activation_RELU_5	-	-	-	(3, 4, 3, 128)	0
Convolution_3D_6	128	(2, 2, 2)	(1, 1, 1)	(3, 4, 3, 128)	131200
Activation_RELU_6	-	-	-	(3, 4, 3, 128)	0
Convolution_3D_7	256	(2, 2, 2)	(1, 1, 1)	(3, 4, 3, 256)	262400
Activation_RELU_7	-	-	-	(3, 4, 3, 256)	0
Convolution_3D_8	256	(2, 2, 2)	(1, 1, 1)	(3, 4, 3, 256)	524544
Activation_RELU_8	-	-	-	(3, 4, 3, 256)	0
Max_Pool_3D_3	-	(1, 1, 1)	-	(3, 4, 3, 256)	0
Convolution_3D_9	512	(1, 1, 1)	(1, 1, 1)	(3, 4, 3, 512)	131584
Activation_RELU_9	-	-	-	(3, 4, 3, 512)	0
Convolution_3D_10	512	(1, 1, 1)	(1, 1, 1)	(3, 4, 3, 512)	262656
Activation_RELU_10	-	-	-	(3, 4, 3, 512)	0
Convolution_3D_11	1024	(1, 1, 1)	(1, 1, 1)	(3, 4, 3, 1024)	525312
Activation_RELU_11	-	-	-	(3, 4, 3, 1024)	0
Convolution_3D_12	2048	(1, 1, 1)	(1, 1, 1)	(3, 4, 3, 2048)	2099200
Activation_RELU_12	-	-	-	(3, 4, 3, 2048)	0
Max_Pool_3D_4	-	(1, 1, 1)	-	(3, 4, 3, 2048)	0
Dropout_0.2_1	-	-	-	(3, 4, 3, 2048)	0
Flatten	-	-	-	(73728)	0
Dense	512	-	-	512	37749248
Activation_Sigmoid_13	-	-	-	512	0
Dropout_0.5_2	-	-	-	512	0
Dense_Output_Softmax	2	-	-	2	1026

**Table 6 diagnostics-10-00402-t006:** Performance evaluation of the 3D CNN model.

Metrics	Split 1	Split 2	Split 3	Split 4	Split 5
Accuracy	93.24%	90.21%	92.68%	95.29%	92.33%
Precision	0.9270	0.9340	0.9520	0.9270	0.9100
Specificity	0.9112	0.8843	0.9100	0.90	0.9263
Recall	0.9140	0.9157	0.9310	0.9430	0.9240
F1-Score	0.9366	0.9290	0.9428	0.9360	0.9160
ROC-AUC	0.94	0.92	0.95	0.98	0.94

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
