# Peer review of "Detection of Parkinson’s Disease from 3T T1 Weighted MRI Scans Using 3D Convolutional Neural Network"

_diagnostics, 2020, doi:10.3390/diagnostics10060402_

Round 1
Reviewer 1 Report
The manuscript is written systematically, based on a well documented methodology increasing the diagnostic possibility of Parkinson's disease.
Author Response
We would like to thank the Editor and reviewers for their in-depth reviews and constructive suggestions, which have substantially improved the quality of the manuscript.
Reviewer 2 Report
The manuscript entitled Detection of Parkinson’s Disease from 3T T1 Weighted MRI scans using 3D Convolutional Neural Network by Sabyasachi Chakraborty, Satyabrata Aich, and Hee-Cheol Kim describes the development of a Neural Network to process and identify PD based on the analysis of 3D MRI scans.
The manuscript is very interesting and well suited for publication in Diagnostics
Below, some suggestions and minor corrections proposed
1 - The authors should use the terms Accuracy, Sensitivity, Specificity and ROC when discussing results and evaluation of 3D CNN.
2 – On Figure 2: Label MRI orientation (sagittal, coronal, etc)
3 – On Figure 3, please include a legend
4 – On Figure 7: Analysing the quantile-quantile plot, why some samples display a prediction probability of 0.8 to 0.85? Similarly, why HC samples have lower prediction probability (around 0.65 - 0.75).
5 – On Figure 8, please include regions label, such as SN, cerebellum, etc…
Minor orthographical corrections:
- Related Work – define HC and MCI.
- Data Collection and Preprocessing
3.1. Data Collection - Out of XXX patients, XXX was female and XXX was male. The scans that were considered for the study belonged to the subjects aged 62.64 ± 9.9. The scans primarily belonged to two research groups that are Healthy Control (HC) and Parkinson’s Disease (PD). The scans were distributed into the respective research groups as XXX for Healthy Control and XXX for Parkinson’s Disease. - Discussion
The study presented in the papers concerns
Author Response
We would like to thank the Editor and reviewers for their in-depth reviews and constructive suggestions, which have substantially improved the quality of the manuscript. The pointwise replies/modifications incorporated in the revised manuscript are mentioned below.
Reply to the Comments:
Comment 1: The authors should use the terms Accuracy, Sensitivity, Specificity and ROC when discussing results and evaluation of 3D CNN.
Reply: In the present work we have used the following Performance Metric
- Accuracy
- Recall; which is also known as Sensitivity
- Precision
- Confusion Matrix
- Specificity (It was previously not added in the study but has been added in the revised version)
- ROC-AUC (It was previously not added in the study but has been added in the revised version)
Comment 2: On Figure 2: Label MRI orientation (sagittal, coronal, etc)
Reply: The MRI orientations labels has been added.
Comment 3: On Figure 3, please include a legend
Reply: The Legend for the Figure 3 was missed out in the previous version but has been added in the present version.
Comment 4: On Figure 7: Analysing the quantile-quantile plot, why some samples display a prediction probability of 0.8 to 0.85? Similarly, why HC samples have lower prediction probability (around 0.65 - 0.75).
Reply: As this work is primarily based upon the Binary Classification, therefore, probability above 0.51 becomes the deciding factor for a particular prediction. Therefore, based on this rule the Confidence or the Probability are converted into the prediction. Now for the PD class some confidence probabilities are 0.8 to 0.85, therefore it is certainly acceptable for the doctor or the physicians to understand the risk level for Parkinson’s Disease. Further such Confidence values can be utilized to develop a Risk Model of Parkinson’s disease based on the stages.
Some HC samples have been found to have lower probability that is around 0.65-0.75. The reason behind this is our model has been developed in such a way that it is more weighted towards the Parkinson’s Class which is mentioned in our Hypothesis. And also we performed an analysis of the Mispredicted HC samples and HC samples of lower confidence i.e 0.60 to 0.70. And it was observed that the Healthy Controls at that period of time are approaching towards the Prodromal Stage. Also, Lower HC confidence also suggests the Risk of a person to develop Parkinson’s Disease in near future.
Comment 5: On Figure 8, please include regions label, such as SN, cerebellum, etc
Reply: The region labels have been added.
Comment 6: Related Work – define HC and MCI.
Reply: Abbreviations have been defined.
Comment 7: Data Collection and Preprocessing
3.1. Data Collection - Out of XXX patients, XXX was female and XXX was male. The scans that were considered for the study belonged to the subjects aged 62.64 ± 9.9. The scans primarily belonged to two research groups that are Healthy Control (HC) and Parkinson’s Disease (PD). The scans were distributed into the respective research groups as XXX for Healthy Control and XXX for Parkinson’s Disease.
Reply: The values were omitted in the previous version but has been added in the present version.
Comment 8: Discussion
The study presented in the papers concerns
Reply: The change has been made.